Identification of four hub genes associated with adrenocortical carcinoma progression by WGCNA

Xia Wang-Xiao 1 2 3 4
Yu Qin 1 2 3 4
Li Gong-Hua 1 2 3
Liu Yao-Wen 1 2 3
Xiao Fu-Hui 1 2 3
Yang Li-Qin 1 2 3
http://orcid.org/0000-0002-4996-3655 Rahman Zia Ur 1 2 3 4
Wang Hao-Tian 1 2 3 4
Kong Qing-Peng 1 2 3 kongqp@mail.kiz.ac.cn
1 State Key Laboratory of Genetic Resources and Evolution/Key Laboratory of Healthy Aging Research of Yunnan Province, Kunming Institute of Zoology, Chinese Academy of Sciences , Kunming , China
2 Center for Excellence in Animal Evolution and Genetics, Chinese Academy of Sciences , Kunming , China
3 Kunming Key Laboratory of Healthy Aging Study , Kunming , China
4 Kunming College of Life Science, University of Chinese Academy of Sciences , Beijing , China
Tao Shi-Cong
Electronic publication date: 2019 Mar 14
Publication date: 2019
Volume: 7
Electronic Location ID: e6555
Received 2018 Oct 25; Accepted 2019 Feb 2
Copyright: © 2019 Xia et al.
Copyright year: 2019
Copyright holder: Xia et al.
License: This is an open access article distributed under the terms of the Creative Commons Attribution License, which permits unrestricted use, distribution, reproduction and adaptation in any medium and for any purpose provided that it is properly attributed. For attribution, the original author(s), title, publication source (PeerJ) and either DOI or URL of the article must be cited.
License URL: https://creativecommons.org/licenses/by/4.0/

Keywords: ACC, WGCNA, Hub genes, Progression

Funding: National Natural Science Foundation of China 81602346 Applied Basic Research Projects of Yunnan Province 2018FB137 Chinese Academy of Sciences QYZDB-SSW-SMC020 and KJZD-EW-L14 This study was supported by grants from the National Natural Science Foundation of China (81602346), Applied Basic Research Projects of Yunnan Province (2018FB137), Chinese Academy of Sciences (QYZDB-SSW-SMC020 and KJZD-EW-L14). The funders had no role in study design, data collection and analysis, decision to publish, or preparation of the manuscript.

==============================
Background

Adrenocortical carcinoma (ACC) is a rare and aggressive malignant cancer in the adrenal cortex with poor prognosis. Though previous research has attempted to elucidate the progression of ACC, its molecular mechanism remains poorly understood.

Methods

Gene transcripts per million (TPM) data were downloaded from the UCSC Xena database, which included ACC (The Cancer Genome Atlas, n = 77) and normal samples (Genotype Tissue Expression, n = 128). We used weighted gene co-expression network analysis to identify gene connections. Overall survival (OS) was determined using the univariate Cox model. A protein–protein interaction (PPI) network was constructed by the search tool for the retrieval of interacting genes.

Results

To determine the critical genes involved in ACC progression, we obtained 2,953 significantly differentially expressed genes and nine modules. Among them, the blue module demonstrated significant correlation with the “Stage” of ACC. Enrichment analysis revealed that genes in the blue module were mainly enriched in cell division, cell cycle, and DNA replication. Combined with the PPI and co-expression networks, we identified four hub genes (i.e., TOP2A, TTK, CHEK1, and CENPA) that were highly expressed in ACC and negatively correlated with OS. Thus, these identified genes may play important roles in the progression of ACC and serve as potential biomarkers for future diagnosis.

Introduction

Adrenocortical carcinoma (ACC) is a rare and aggressive malignant cancer found in the adrenal cortex (Fay et al., 2014). While this disease can occur at any age, it tends to show a bi-modal distribution with an initial peak in childhood (1–6 years old) and a second peak in middle-age (40–50 years old) (Kiseljak-Vassiliades et al., 2018). As ACC has no obvious phenotypic traits at the early stage, almost 70% of patients are at stage III or IV when diagnosed (Bharwani et al., 2011; Fay et al., 2014). At these stages, ACC is invasive and metastatic, with patients at stage IV only having a 5-year survival of 6–13% (Else et al., 2014; Fassnacht et al., 2009; Fassnacht, Kroiss & Allolio, 2013). Unfortunately, current ACC therapies, such as surgery, chemotherapy, and radiotherapy, exhibit poor performance and outcomes (Allolio & Fassnacht, 2006). While next generation sequencing technology recently identified several genetic molecules associated with ACC (Soon et al., 2008; Assié et al., 2014; Greenhill, 2016; Zheng et al., 2016; Chortis et al., 2018), our understanding of ACC progression at each stage remains incomplete and treatment options are limited (Hoang, Ayala & Albores-Saavedra, 2002; Cherradi, 2016). Thus, integrated analysis is required to further understand the molecular characterization of ACC gene expression, which may indicate stage and identify additional biomarkers for further research and clinical therapies.

Traditional methods of identifying the functional genes of ACC have focused on screening differentially expressed genes (DEGs) (Giordano et al., 2003; Slater et al., 2006; Lombardi et al., 2006), with limited attention paid to gene connections. Weighted gene co-expression network analysis (WGCNA) is a popular method in systems biology that can construct gene networks and detect gene modules (Clarke et al., 2013; Yang et al., 2014; Lee et al., 2015; Goldman et al., 2017; Sun et al., 2017). By analyzing the connectivity between modules and clinical traits, we can determine which modules are associated with which traits. Those genes found in the center of a regulation network usually exhibit more important functions. Thus, the degree of gene connectivity in one module can also be analyzed by the gene-gene interaction/regulation network, from which critical hub genes can be identified.

In this study, we identified genes involved in ACC progression via comprehensive transcriptome-wide analysis of ACC gene expression patterns. We systematically analyzed clusters of genes with similar expression patterns using WGCNA and found the MEblue module to be highly related to clinical stage. Further analysis identified four hub genes (i.e., TOP2A, TTK, CHEK1, and CENPA) from the module that were associated with ACC progression and prognosis. Thus, these hub genes may serve as candidate biomarkers of ACC in clinical treatment and contribute to a greater understanding of ACC progression.

Materials and Methods

Data collection

We obtained gene expression transcripts per million (TPM) values (Table S1) from the UCSC Xena (http://xena.ucsc.edu/) database, which included 77 ACC samples from the cancer genome atlas (TCGA) (https://cancergenome.nih.gov/) and 128 normal samples from genotype tissue expression (GTEx) (https://www.gtexportal.org/home/). The two databases raw sequencing reads were recalculated with a unifying pipeline. Clinical data were downloaded from TCGA using the “cgdsr” package in R (v3.1.3) (R Core Team, 2015; Jacobsen, 2015).

DEG screening

Of the 60,498 genes in each sample, we removed genes with a mean TPM ≤ 2.5 (>1 is a common cutoff for determining if an isoform is expressed or not (Liu, Jing & Tu, 2016)) in the cancer and normal samples and thus retained 13,987 genes. For those genes in the samples that showed significant changes, we used analysis of variance (ANOVA) in R (R Core Team, 2013) to determine the variance in genes between the two groups. ANOVA is a collection of statistical models useful for DEG analysis (Alabi et al., 2018; Monterisi et al., 2015). We obtained 2,953 significant DEGs (Table S2) in ACC with a p < 0.001 and |log2 (fold-change)| > 1 cutoff.

Co-expression network construction by WGCNA

Weighted gene co-expression network analysis (v1.49) can be applied to identify global gene expression profiles as well as co-expressed genes. Therefore, we installed WGCNA package for co-expression analysis using Bioconductor (http://bioconductor.org/biocLite.R). We used the soft threshold method for Pearson correlation analysis of the expression profiles to determine the connection strengths between two transcripts to construct a weighted network. Average linkage hierarchical clustering was carried out to group transcripts based on topological overlap dissimilarity in network connection strengths. To obtain the correct module number and clarify gene interaction, we set the restricted minimum gene number to 30 for each module and used a threshold of 0.25 to merge the similar modules (see the detailed R script in File S1).

Identification of clinically significant modules

We used two methods to identify modules related to clinical progression traits. Module eigengenes (MEs) are the major component for principal component analysis of genes in a module with the same expression profile. Thus, we analyzed the relationship between MEs and clinical traits and identified the relevant modules. We used log10 to transform the p-value from the linear regression between gene expression and clinical stage, which was defined as gene significance. Average gene significance in a module was defined as module significance.

Functional and pathway enrichment analysis

The database for annotation visualization and integrated discovery (DAVID) (v6.8) (http://david.abcc.ncifcrf.gov/) was used for functional annotation of genes to better understand their biological functions. All genes in the blue module were uploaded for gene ontology (GO) and Kyoto encyclopedia of genes and genomes (KEGG) pathway enrichment analyses, with cutoffs of p < 0.01 and p < 0.05 established for significant biological processes and pathways, respectively.

PPI and co-expression analysis

Genes were uploaded to the search tool for the retrieval of interacting genes (STRING) (v10.5) (https://string-db.org/) database. Confidence was set to more than 0.4 and other parameters were set to default. We visualized the gene co-expression network with Cytoscape (v2.7.0) (Shannon et al., 2003).

Gene expression correlation with stage and survival analysis

The correlation between gene expression and stage was determined using GEPIA (http://gepia.cancer-pku.cn/index.html) (Tang et al., 2017). The correlation between gene expression and overall survival (OS) was established using the Cox model. A hazard ratio p-value of <0.01 was considered significant. Each gene with higher expression in the ACC samples had corresponding lower survival expectation. The “limma” (Ritchie et al., 2015) R package was used to test for the significantly expressed gene in GSE10927.

Results

Construction and analysis of gene co-expression network with DEGs in ACC

Genes with a mean TPM ≤ 2.5 were removed from the two groups and the remaining 13,987 genes were used for differential expression analysis with ANOVA. In total, 2,953 significant DEGs were identified with a cutoff of p < 0.001 and |log2(fold-change)| > 1 (Fig. 1A), which included 1,181 up-regulated and 1,772 down-regulated genes (Fig. 1B). The 2,953 gene expression levels in ACC and normal samples are shown in the heatmap in Fig. 1C and Table S2.

Figure 1 Nine modules obtained following WGCNA analysis of DEGs in ACC.

(A) X-axis represents log2 fold-changes and Y-axis represents negative logarithm to the base 10 of the p-values. Black vertical and horizontal dashed lines reflect filtering criteria (FC = ±1 and p-value = 0.001). (B) Red and blue bars are number of significantly down-regulated (n = 1,772) or up-regulated genes (n = 1,181) in ACC compared with non-tumor samples. (C) Heatmap shows all DEGs in ACC and GTEx. The Log2(TPM + 0.001) expression level of each gene profile from each sample is represented by color. (D) Sample clustering was conducted to detect outliers. This analysis was based on the expression data of DEGs between tumor and non-tumor samples in ACC. All samples are located in the clusters and pass the cutoff thresholds. Color intensity is proportional to sample age, gender, status, and stage. (E, F) Soft-thresholding power analysis was used to obtain the scale-free fit index of network topology. (G) Scale free topology when β = 6. (H) Hierarchical cluster analysis was conducted to detect co-expression clusters with corresponding color assignments. Each color represents a module in the constructed gene co-expression network by WGCNA. (I) Heatmap depicts the Topological Overlap Matrix (TOM) among 500 randomly selected genes from the DEG weighted co-expression network. Light color represents lower overlap and red represents higher overlap.

Genes with similar expression patterns may participate in similar biological processes or networks (Mao et al., 2009). To better understand the gene expression network during ACC development, the co-expression network of the 2,953 DEGs was analyzed by WGCNA. First, to determine whether all 77 ACC samples were suitable for network analysis, the sample dendrogram and corresponding clinical traits were analyzed. We found that all samples were included in the clusters and passed the cutoff thresholds (Fig. 1D). The power value is a critical parameter that can affect the independence and average connectivity degree of the co-expression modules. Thus, network topology using different soft thresholding powers was screened, with β = 6 (scale free R2 = 0.928) selected for later analysis (Figs. 1E, 1F and 1G). We then constructed the gene co-expression network using WGCNA based on the hierarchical clustering of the calculated dissimilarities, and nine modules were obtained (Fig. 1H; Table S3). We used eigengenes as representative profiles and quantified module similarity by eigengene correlation (Fig. 1I).

Correlation of blue module with clinical stage and progression

We investigated whether any module was correlated with clinical stage and tested the relevance between each module and ACC clinical traits. We found that module significance of the blue module was higher than that of any other, implying it had greater correlation with ACC stage (Fig. 2A). The blue module also displayed a positive correlation with ACC clinical stage (r = 0.5, p = 6e-06) and negative correlation with OS (r = −0.56, p = 3e-07) (Fig. 2B). The eigengene dendrogram and heatmap indicated that the MEblue and MEyellow modules were highly correlated with clinical stage (Fig. 2C). Finally, gene significance and module membership were plotted for the blue module (Fig. 2D), which indicated that this module was significantly related to clinical stage.

Figure 2 Correlation of Blue module with clinical stage.

(A) Bar plot of mean gene significance across genes associated with ACC stage in the module. (B) Heat map with each cell containing the p-value correlation from the linear mixed-effects model. Row corresponds to module; column corresponds to ACC clinical traits. Results indicate that MEblue is highly related to patient stage. (C) The dendrogram shows the relation of modules with stage and the heatmap shows the eigengene adjacency. (D) Correlation between MEblue membership and gene significance. (E) GO enrichment analysis of 650 genes in MEblue identified biological processes related to cell proliferation. Y-axis represents significance of enrichment results transformed to “−log(p-value).” (F) KEGG enrichment analysis of 650 genes in MEblue identified pathways related to cell cycle and DNA replication.

To determine the function of the 650 genes in the blue module, GO and KEGG function and pathway enrichment analyses were performed by DAVID functional annotation (Huang, Sherman & Lempicki, 2009). For GO biological processes, genes in the module were significantly enriched in cell division (p = 1.05e-26) (Fig. 2E; Table S4), whereas for KEGG pathway analysis, the genes were mainly enriched in cell cycle (p = 2.7e-19) and DNA replication (p = 8.27e-8) (Fig. 2F; Table S5) pathways. These processes and pathways all play critical roles in cancer progression (Tachibana, Gonzalez & Coleman, 2005), implying that genes in this module may be involved in ACC progression.

PPI and co-expression networks to identify hub genes in ACC progression

To clarify high confidence hub genes, we entered the blue module genes into the STRING (Szklarczyk et al., 2015) database. The genes were ranked by the protein–protein interaction nodes and the top 5% of genes (16 genes) were chosen as candidate hub genes (Fig. 3A; Table S6). As highly connected hub genes in a module play important roles in biological processes (Liu, Jing & Tu, 2016), genes in the blue module were ranked by their degree of gene co-expression connectivity (Table S7). To identify genes that may play notable roles in ACC progression, the top 5% of genes (31 genes) (Fig. 3B) in the blue module with the highest connectivity were classified as candidate hub genes for further analysis. Finally, four common genes (i.e., TOP2A, TTK, CHEK1, and CENPA) in the two analysis were identified as hub genes in ACC (Fig. 3C). These four genes were highly expressed in ACC samples compared with normal samples (Figs. 3D–3G), indicating that they likely act as oncogenes in ACC. Further analysis of the GSE10927 dataset, which included microarray data of 10 normal samples and 33 ACC samples (Human Genome U133A 2.0 Plus; Affymetrix, Santa Clara, CA, USA) (Giordano et al., 2009), demonstrated that the four genes showed significant high expression in ACC (Figs. S1A–S1D). Furthermore, based on immunoreactivity experiments, TOP2A is reported to be highly expressed in ACC (Giordano et al., 2003).

Figure 3 Four hub genes identified through PPI and gene-gene connection network.

(A) PPI network of genes in MEblue. Intersection of top 50 genes in MEblue is shown, red nodes are hub genes of the network. (B) Co-expression network of top 50 genes in MEblue, red nodes are hub genes of the network. (C) Venn diagram shows common hub genes between co-expression and PPI network analyses. (D–G) Four hub genes significantly expressed in ACC samples compared with corresponding GTEx tissue samples (*p < 0.001).

Significant associations of hub genes with ACC stage and survival

We investigated the four hub genes to better understand their functions. We found that TOP2A, TTK, CHEK1, and CENPA play critical roles in biological processes that are highly correlated with cancer (Dominguez-Brauer et al., 2015), such as DNA topological structure, cell cycle progression, and mitosis (Hoffmann et al., 2011; Liu et al., 2000; De Resende et al., 2013; Thu et al., 2018), thereby suggesting their possible role in cancer development. Further exploration of their expression patterns during ACC clinical progression showed that the levels of these genes were significantly altered with clinical stage and markedly increased at stage III and IV (Figs. 4A–4D). This correlation between the expression levels of the four genes and ACC progression may be useful in ACC diagnosis.

Figure 4 Significant correlation between hub gene expression with pathological stage and survival.

(A–D) Significant correlation between expression levels of TOP2A, TTK, CHEK1, and CENPA with ACC pathological stage. (E–H) Survival plot of OS in ACC. Higher expression (red line) of TOP2A, TTK, CHEK1, and CENPA indicates poorer prognosis. HR, hazard ratio.

Tumor prognosis is an important feature in cancer and has attracted considerable attention. To assess the utility of WGCNA at identifying hub genes indicative of ACC, we conducted survival analysis (Figs. 4E–4H). We separated the samples into two groups according to median gene expression levels and performed survival analysis using the Cox model. Survival analysis showed that the expression of all four genes was significantly correlated with OS (Figs. 4E–4H), with higher expression associated with lower patient survival time. The correlation between the hub genes and ACC prognosis suggests that these four genes likely contribute to the progression of ACC.

Discussion

As ACC exhibits no obvious phenotypic traits during its early stages, diagnosis is often delayed in many patients (Bharwani et al., 2011; Fay et al., 2014). We systematically analyzed gene expression and found potential biomarker genes for ACC diagnosis. To identify genes that may play central roles in ACC progression, gene co-expression network analysis was conducted using WGCNA, which can describe correlation patterns among genes at the RNA level. Based on WGCNA, we obtained nine modules, with each module containing an average of 217 genes. Only 205 genes were unclassified in any module (in gray), accounting for 10.50% of DEGs. In comparison, previous studies have reported an average gene number in each module of 216 to 336 and percentage of genes not found in any module of 5.67–33.61% of DEGs (Liu et al., 2017, 2018; Yang et al., 2018; Zuo et al., 2018). In conclusion, our WGCNA results were comparable. We identified four hub genes (i.e., TOP2A, TTK, CHEK1, and CENPA) in the network center related to gene regulation and possible carcinogenesis.

Genes located in the central position of a gene-gene interaction network likely exhibit more important functions than other genes. Further investigation found that these four hub genes contribute to several tumor types indeed. For instance, TOP2A (topoisomerase II alpha), a specific marker of cell proliferation, is the primary molecular target of anthracyclines used for treating breast cancer (Villman et al., 2006; Wang et al., 2012). TTK, also known as monopolar spindle 1 (MPS1), plays a key role in cancer cell growth and proliferation, with its inhibition able to decrease tumor aggressiveness (Al-Ejeh et al., 2014; Maire et al., 2015; Zhu et al., 2018). CHEK1 (checkpoint kinase 1), a conserved serine/threonine kinase, plays a key role in tumor growth promotion (Zhang & Hunter, 2013). Furthermore, inhibition of CHEK1 expression by UCN-01, CEP-3891 (Zhu et al., 2018), AZD7762, or LY2606368 inhibitors (Manic et al., 2017) can prevent the proliferation of cancer cells (Bryant, Rawlinson & Massey, 2014; Schuler et al., 2017). CENPA (centromere protein A), a histone H3 variant, is highly expressed in cancers, including breast, colorectal, liver, lung, ovarian, and osteosarcoma (Athwal et al., 2015; Sun et al., 2016; Filipescu et al., 2017). In addition, inhibition of CENPA expression in cancer cells can reduce sphere forming ability, proliferation, and cell viability (Behnan et al., 2016). Here, our study revealed that the expression levels of all four hub genes were significantly correlated with ACC progression (Figs. 4A–4D) and OS (Figs. 4E–4H), suggesting their critical function in ACC. Our results indicated that these four genes may play key roles in ACC tumorigenesis. However, the specific functions of these genes that contribute to ACC cell proliferation, differentiation, and metastasis need further study.

Conclusions

Based on gene co-expression network analysis, we identified four hub genes that likely contribute to the progression of ACC. The expressions of the four hub genes demonstrated significant correlation with ACC clinical stage and prognosis (Figs. 4A–4H). Thus, these four genes may act as potential biomarkers in predicting clinical outcomes and diagnosis of ACC. Furthermore, inhibitors of TOP2A, TTK, and CHEK1, which are already used for treating certain cancers, could potentially be used in ACC treatment. Further experimental and clinical studies are required to extend these findings.

Supplemental Information

Supplemental Information 1 The four hub genes significantly expressed in GSE10927 dataset.

(A-D) The four hub genes significantly expressed in ACC samples compared with normal tissue samples in GSE10927 dataset.

Click here for additional data file.

Supplemental Information 2 The 77 cancer and 128 normal samples id.

Click here for additional data file.

Supplemental Information 3 The log2(TPM+0.001) of 2953 DEGs.

Click here for additional data file.

Supplemental Information 4 Genes in each module.

Click here for additional data file.

Supplemental Information 5 Go enrichment analyses of 650 genes in the blue module.

Click here for additional data file.

Supplemental Information 6 Kegg pathway analysis of 650 genes in the blue module.

Click here for additional data file.

Supplemental Information 7 The 650 gene protein protein interaction analysis.

Click here for additional data file.

Supplemental Information 8 The 650 gene co-expression connectivity degree.

Click here for additional data file.

Supplemental Information 9 R script of WGCNA analysis.

Click here for additional data file.

We thank Qiong-Hua Gao for suggestions in modifying the paper and Christine Watts for help in honing the manuscript.

Additional Information and Declarations

Competing Interests

Author Contributions

Data Availability

The authors declare that they have no competing interests.

Wang-Xiao Xia conceived and designed the experiments, analyzed the data, prepared figures and/or tables, authored or reviewed drafts of the paper.

Qin Yu authored or reviewed drafts of the paper.

Gong-Hua Li analyzed the data, authored or reviewed drafts of the paper.

Yao-Wen Liu authored or reviewed drafts of the paper.

Fu-Hui Xiao authored or reviewed drafts of the paper.

Li-Qin Yang contributed reagents/materials/analysis tools.

Zia Ur Rahman authored or reviewed drafts of the paper.

Hao-Tian Wang prepared figures and/or tables.

Qing-Peng Kong conceived and designed the experiments, authored or reviewed drafts of the paper, approved the final draft.

The following information was supplied regarding data availability:

The raw measurements are available in the Supplemental Files.

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
