# Peer review of "Identification of four hub genes associated with adrenocortical carcinoma progression by WGCNA"

_PeerJ, doi:10.7717/peerj.6555_

## Round 0.1 · original submission · Major Revisions

Although it is of interest, we are unable to consider it for publication in its current form. Many improvements are still needed in the current version. The reviewers have raised a number of points which we believe would improve the manuscript. If you are able to fully address these points, we would encourage you to submit a revised manuscript. Please include a point-by-point response.

Reviewer 1 ·

Basic reporting

no comment

Experimental design

In order to replicability of the related results of the study, R scripts should be uploaded as a supplement file.

Validity of the findings

The conclusion is welcome, but should be verified by molecular biology experiment such as quantitative real-time PCR, western blot. Otherwise, the result is not persuasive.

Additional comments

Authors used public databases (TCGA and GTEx) to identify four hub genes (TOP2A, TTK, CHEK1, and CENPA) associated with adrenocortical carcinoma progression by weighted gene co-expression network analysis. However, some questions needs to be pointed out.
1. The manuscript needs polishing, which contributes to improve readability of the manuscript.
2. Results of the present study were only based on mRNA level, lacked confirmatory experiments including quantitative real-time PCR, western blot. Thus, the current result was not persuasive.
3. In order to replicability of the related results of the study, R scripts should be uploaded as a supplement file. Furthermore, version numbers of bioinformatics tools should be annotated.

·

Basic reporting

The overall structure of the manuscript is well-organized, but there are several issues authors should address.
1. Figure 1A legend: “fold-changes” should be replaced with “log2 fold-changes”. “lines” should be replaced with “dashed lines”.
2. Figure 1C: the heat map should be better labeled.
3. Figure 2: the sequence of each sub-panel is inconsistent with the other figures.
4. Supplemental Table 1: Sheet 1 and 2 should be better labeled.
5. Some references in the introduction are not appropriate. Line 45 – 46: Authors should use primary sources. Line 47: Authors’ claim is not substantiated in the referenced article. Line 56: Referenced article is on small non-coding RNAs and it should not be seen as the traditional method of identifying functional genes.
6. Language should be improved for easier comprehension. Examples include Lines 41-42, 46-47, 86-88, 198-200, where the current phrasing is difficult to understand.

Experimental design

In general, the rationale and methodology seem appropriate, but there are still some issues.
1. Authors did not clarify what specific types of data they obtained from TCGA and GTEx (Level 1 to Level 3).
2. If Level 3 data was obtained, the gene expression level from TCGA and GTEx cohorts would not be directly comparable, due to the use of different analysis pipelines. It is necessary to have a unifying pipeline in analyzing the raw sequencing data from the two cohorts (Qingguo et al. https://www.nature.com/articles/sdata201861).
3. The method authors used to identify differentially expressed genes is not very robust. I recommend a validation of DEGs using EdgeR (http://bioconductor.org/packages/edgeR/) or DESeq2 (http://bioconductor.org/packages/DESeq2/). Detailed information can be seen in the relevant papers.

Validity of the findings

The conclusion authors drew is based on the data they presented. However, the validation of the data is necessary.
1. I recommend qPCR validation of the 4 hub genes identified in ACC cell lines. Of course, analyzing human primary samples would be more satisfactory.
2. It will also be helpful if authors could analyze data from other public datasets to confirm the differential expression of the 4 hub genes.

Additional comments

The manuscript is a computational analysis of RNA-seq data in ACC, using tumor samples from TCGA and normal samples from GTEx. Authors described (i) a list of differentially expressed genes in ACC vs normal samples, (ii) 9 gene modules identified by WGCNA, (iii) 4 hub genes associated with ACC progression. Overall, the manuscript is well-organized in a professional manner and provides novel aspect in the progression of ACC. However, a number of issues about the writing and methods (as mentioned above) must be addressed before acceptance.

Reviewer 3 ·

Basic reporting

Adrenocortical carcinoma is a vicious and aggressive cancer, so every new study is quite important.
The work done by the researchers here is particularly interesting and well written. The reader easily follows the progress of the study and I find that the whole is very educational. The data is made available, so it's a very successful search.
The introduction is well balanced highlighting the crucial data of the field, without being too boring to read.
The materials and methods are correct, but it would be good to highlight the choice of different parameters.
Results section is well designed and I like to follow each step of the study.
Unfortunately, I find the figures difficult to read.
Figures 1C, 1G and 1H for example are unusable. Similarly, the figures of Fig 2B, the labels of 2E and 3B are too small. It would be good to improve them.

Experimental design

The methodology used is correct and interesting.
An important point is to discuss the various parameters used.
Thus, I see why 2.5 as TPM, but it would be more detailed this choice for the reader, also estimate (give) the implication that would have given a choice with a stronger value (or weaker).
Similarly, the choice of aov must be discussed.
An underlying question is the sensitivity of modules to the choice of DEGs.

Validity of the findings

As said, "most are expressed at extremely low levels", but is it also experimentation (laboratory) dependent? Variations in the provenance of these data must be discussed. It is unclear whether standardization for example is general or by initial experience.
"Average linkage hierarchical" is a classic approach, but would not Ward have been more relevant?

“KEGG pathway analysis, the genes were mainly enriched in cell cycle and DNA replication (Fig. 2F, Table S5) These processes and pathways all play critical roles in cancer progression (Tachibana et al., 2015), implying that genes in this module may be involved in ACC progression. " Can you be more specific and tell us how to achieve this result quantitatively. What statistical test and under which hypothesis can you conclude?

"The top 5% of genes (16 genes) with confidence" were chosen as "hubs for PPI analysis" (Fig. 3A, Table S6). But how is this choice more relevant than 10% or 1%? Similarly, if you remove one of these hubs, is the set still stable or is it terribly sensitive to given hubs?

Additional comments

As a reviewer, I like reading articles that allow me to follow the path of the authors, which is perfectly the case. There is a real choice to search for genes in a supervised way, and I trust the results found. However, it lacks a little criticism on the choices made and especially the influence of the parameters. These corrected points will make this article an excellent article.

---

## Round 0.2 · Major Revisions

As you can see, while only one of the original reviewers was able to re-review the revised manuscript, several other reviewers stepped in and have provided valuable comments, in particular the comments of Rev 6.

Please address their comments in a suitable revision.

Reviewer 3 ·

Basic reporting

There are literature references with sufficient field background/context provided; the article is professionally done with appropriate figs, tables and sufficient data and script shared. It is self-contained with relevant results to hypotheses.

Experimental design

it is rigorous and well done, it can be reproduced if needed.

Validity of the findings

in regards to state of art, it is robust and statistically sound and controlled.

Additional comments

The author answered all my questions and we incorporated these modifications to the manuscript, which can be published from my point of view as it stands.

Reviewer 4 ·

Basic reporting

The work is very much interesting one. I have found that majority of queries in my mind already raised by three reviewers and authors already provided their responses which are quite satisfactory. So, present form can be accepted now.

Experimental design

Satisfactory

Validity of the findings

Acceptable

Additional comments

The work is very much interesting one. I have found that majority of queries in my mind already raised by three reviewers and authors already provided their responses which are quite satisfactory. So, present form can be accepted now.

·

Basic reporting

Clear and unambiguous, professional English used throughout.

Experimental design

The study is well designed, carried out properly, and technically and statistically sound.

Validity of the findings

The results are comprehensible , conclusive and reliable.

Additional comments

By weighted gene correlation network analysis (WGCNA), Wang-Xiao Xia. et al attempted to identified four hub genes associated with adrenocortical carcinoma progression. They found that the hub genes (TOP2A, TTK, CHEK1, and CENPA) were highly expressed in ACC and negatively correlated with OS. Thus, these identified genes may play important roles in the progression of ACC and serve as potential biomarkers for future diagnosis. The study is well designed, carried out properly, and technically and statistically sound. The results are comprehensible , conclusive and reliable.

Reviewer 6 ·

Basic reporting

The authors use an approache based purely on mRNA levels to identify genes related with ACC. They perform differential expression assuming that the Xena database (the repository from which the case/control datasets were obtained) correctly armonizes mRNA from two different datasets. Can we assume that? This is crucial for the paper to be credible.

They follow up with co-expression networks (WGCNA methodology) and PPI to identify 4 genes involved in ACC and negatively correlated with survival.

Experimental design

Line 58 systematics biology -> systems biology

What are those GTEx control samples, where do they come from? I mean, version of GTEx gene expression, tissue, age and gender

In line 77 they say ACC and GTEx data sets were integrated into a single pipeline. But there are no details about that one. Since the paper is about detecting differentially expressed/connected genes, this part of the pipeline is precisely the crucial one and should be clearly detailed, at least in a supplementary. The fact that they mention the Xena database is not enough. They should describe how this is done and make the point clear that both datasets can be compare because of that because this approach may be full of different biases: e.g. project, sequencing technology, softwares for producing the TPM quantification… Genes being quantified in the same units (TPM) does not necessarily imply they can be put together. If no enough details are given on how they integrated both data sources, I wouldn’t believe what is found there at least in terms of differential expression. In summary, the paper has to give more details about how the DDBBs are integrated and processed together. It will make the paper much more credible.

The authors claim that WGCNA is a package from Bioconductor but is not, it is from CRAN. Maybe the authors are referring to WGCNA tailored to TGCCA.

Line 147: the significance of the blue module being higher (I suppose the authors are actually saying that the p-value is lowest). However, lower p-values, although they are correlated with the magnitude of the correlation, it does not necessarily mean that correlation with blue module will be the highest. Anyway, the authors should report the correlation in that line. The authors do not report how they calculate gene significance and other parameters. The title of figure 2.A is incomplete, did not mention ACC.

The number of modules is suspiciously low. There are many grey genes and this was not properly addressed in the paper. Genes in the grey module do not have high correlation between them but are there because they shoudn’t be in any of other modules. This should be discussed.

Line 164, what is confidence?




It is not clear the role of PPI in the whole analysis further than the generation of nice plots. What is the conclusion drawn from this particular analysis and how it combines with co-expression networks? This should be either clarified or removed in the case it says nothing about involvement of the genes in ACC progression.

Validity of the findings

Is the paper showing something new? They themselves cite other papers that lead to the same conclusions, i.e. the four genes selected from the blue module are also seen as differentially expressed in other papers. Lines from 207 to 223 reinforce what I am saying as well.

The authors keep saying that the four genes contribute to the progression of ACC but this is not necessarily true. Because one can not affirm whether these genes are upstream or downstream. There is no causality anchor and the genes may be over expressed as a consequence of ACC. So instead of genes contributing to ACC progression, ACC progression may contribute to over expression of the genes. The genes are involved in ACC progression, this is certain but further conclusions are not realistic. The authors should put some effort in finding the causal anchor. And this is only possible when the genetics & environment information come into play.

Additional comments

The paper may be useful for the community but I doubt it is saying something not know already, and there are some pitfalls in the design that should be properly addressed.

·

Basic reporting

The manuscript is well written. Authors handled the gene dataset in an interesting way which would have a clinical impact and more in-depth analysis on the mechanisms underlying different phenotypes.
The abstract is well structured and concise.

Experimental design

Introduction:
First mentioned as abbreviation MEblue module in line 67? (module eigengene E(blue) of the blue module)



Differentiation between the two peaks (childhood and middle age) of the disease in the analysis would be preferential to pick the biological and mechanistic difference between them.

Validity of the findings

"The correlation between gene expression and stage was determined using GEPIA": screen shot please the option of clinical data availability in that link provided.


First 5 lines of result section is repetitive and should be restricted in the methodology.

Fig. 1C is so faint and thus unreadable.
Fig 2C is referred to the info "MEblue and MEyellow modules were highly correlated with clinical stage" and that is not clear in the corresponding figure.

PPI section in the results: line 162-169 is unclear what exactly was done.

Additional comments

No commrnts

---

## Round 0.3 · accepted · Accept

The manuscript has been appropriately improved and reached acceptable quality.

# ·

Basic reporting

Reviewer comments were addressed.

Experimental design

Reviewer comments were addressed.

Validity of the findings

Reviewer comments were addressed.

Additional comments

Reviewer comments were addressed.